**Data Availability Statement:** All relevant data are within the paper and its Supporting information files.

**Funding:** This work was supported by National Science Foundation of China (61773082),

# Asymptotically local synchronization in interdependent networks with unidirectional interlinks

**Zilin Gao**[1☯], **Weimin Luo**[1☯], **Aizhong Shen**[2☯]*

**1** School of Computer Science and Engineering, Chongqing Three Gorges University, Chongqing, China,
**2** Faculty of Professional Finance and Accountancy, Shanghai Business School, Shanghai, China

☯ These authors contributed equally to this work.
* 21190061@sbs.edu.cn

## Abstract

Synchronization in complex networks has been investigated for decades. Due to the particularity of the interlinks between networks, the synchronization in interdependent networks has received increasing interest. Since the interlinks are not always symmetric in interdependent networks, we focus on the synchronization in unidirectional interdependent networks to study the control scheme. The mathematical model is put forward and some factors are taken into consideration, such as different coupling functions and strengths. Firstly, the feasibility of the control scheme is proved theoretically by using Lyapunov stability theory and verified by simulations. Then, we find that the synchronization could be maintained in one sub-network by utilizing our control scheme while the nodes in the other sub-network are in chaos. The result indicates that the influence of interlinks can be decreased and the proposed scheme can guarantee the synchronization in one sub-network at least. Moreover, we also discuss the robust of our control scheme against the cascading failure. The scheme is verified by simulations to be effective while the disturbances occur.

## Introduction

Real-world is composed of large numbers of complex networks. The states of the nodes and couplings change continuously or discretely in a single network and multilayer complex networks [1, 2]. Buldyrev et al. [3] present the concept of interdependent networks in 2010 and discuss the particularity of the interdependent links. The interdependent relations can be found in many real-world network systems, such as social networks in which the same actors are shared [4], plant-communication networks in which the computers work with the support from the power plant and conversely deliver control messages to them [3], transportation networks in which the same locations are shared by airplanes, buses, and trains [5]. Nowadays, interdependent networks has become one of the hot topics in the field of complex networks [6, 7].

The synchronization in complex networks has been concerned for decades. Several synchronization methods have been proposed, such as cluster synchronization [8], phase

Chongqing Postdoctoral Science Foundation (cstc2021jcyj-bshX0035), Chongqing Social Science Planning Project (2021BS038), Chongqing Basic and Advanced Technology Research Project (cstc2018jcyjA2453), and Key Laboratory of Chongqing Municipal Institutions of Higher Education ([2017]3). These funds support us in preparation of the manuscript.

**Competing interests:** The authors have declared that no competing interests exist.

synchronization [9, 10], projective synchronization [11], general synchronization [12, 13] and lag synchronization [14, 15]. Meanwhile, a number of control schemes, include adaptive control [16], decentralized control [17], impulse control [18], and pinning control [19], are provided. In the existing literature, the synchronization in multilayer networks has received increasing interest. The quantity and distribution of the controllers are analyzed to achieve lag synchronization [14]. In Ref. [15], the realizations of different kinds of synchronization with the consideration of different dynamics for each node between general complex networks are discussed. It is found that synchronization could be achieved when driving-response networks have identical connections [20]. To investigate synchronization on complex networks of networks, the attack and robustness of the pinning scheme are analyzed [21].

However, far too little attention has been paid to the synchronization in interdependent networks. The synchronization behavior and the synchronicity in the interdependent systems are discussed in Ref. [22] and [23]. The mathematical models of interdependent networks are proposed and a variety of control schemes are provided to achieve synchronization [24–26]. These studies to date have tended to focus on interdependent networks with bidirectional interlinks, but it is found that the interdependencies are not always mutual or symmetric [27–29]. For instance, a computer hub might work with electricity supplied by a power substation but does not necessarily provide information control to it. Similarly, the operation of a gas station must depend on electricity supplied by a power plant, but the power plant does not need the support of a gas station.

Moreover, the coupling function of the nodes in different sub-networks is usually regarded as the same one and one variable is always used instead of a matrix to describe the intercoupling between sub-networks [22, 24, 26]. These are not appropriate. The couplings are complicated. In [30], the dynamics of the local synchronization is studied in adaptively coupled neuronal network, in which the coupling between two neurons is determined dynamically by the states of the neurons. As for the interdependent networks, both the coupling functions and strengths must be different in different sub-networks at least. And note that the interdependency does not only exist between nodes $i$ in one sub-network but also in the other sub-network. This indicates that the intercoupling matrix is very important and should not be ignored.

The aim of this paper is to explore the realization of synchronization in interdependent networks with unidirectional interlinks. The major contributions of our work are as follows. First, we propose the model of a unidirectional interdependent network composed of two sub-networks. This is different from previous works. Second, different coupling functions and strengths are considered in the proposed model, and an intercoupling matrix is introduced. These are in line with the fact. Third, the design of controller is more simply, and the feasibility of the control scheme is proved theoretically by using Lyapunov stability theory and verified by simulations. Finally, by utilizing our control scheme, the influence of the interdependencies is decreased, and the synchronization could be maintained in one sub-network while the nodes in the other sub-network are in chaos.

The rest of this paper is organized in the following way. In section II, the model of unidirectional interdependent networks is proposed, and some preliminaries are given. The control scheme is presented and proved theoretically in section III. The simulations are run and the results are discussed in section IV. The conclusion is given in section V.

## Model presentation and preliminary

Considering the interdependent networks composed of two sub-networks X and Y. Each sub-network is consisting of nodes which $n$-dimension nonlinear systems and coupled to each

other, and each node is an n-dimension system. Sub-network Y unidirectionally depends on sub-network X with one-to-one mode. Then the dynamic equations of two sub-networks could be described as follows

$$\dot{x}_i = f(x_i) + \alpha \sum_{j=1}^{N} a_{ij}^x H(x_j) \quad i = 1, 2, \ldots, N \tag{1}$$

$$\dot{y}_i = g(y_i) + \beta \sum_{j=1}^{N} a_{ij}^y K(y_j) + \gamma \sum_{j=1}^{N} c_{ij}(H(x_j) - K(y_i)) \quad i = 1, 2, \ldots, N \tag{2}$$

where, for node $i$, $x_i \in \mathbb{R}^n (y_i \in \mathbb{R}^n)$ is the state vector in sub-network X(Y); $f(x_i) : \mathbb{R}^n \to \mathbb{R}^n (g(y_i) : \mathbb{R}^n \to \mathbb{R}^n)$ is a smooth nonlinear vector function; $H : \mathbb{R}^n \to \mathbb{R}^n (K : \mathbb{R}^n \to \mathbb{R}^n)$ is a smooth nonlinear coupling function in sub-network X(Y); $\alpha(\beta)$ is the coupling strength in sub-network X(Y); $\gamma$ is the intercoupling strength from sub-network Y to sub-network X; $A^x = (a_{ij}^x) \in \mathbb{R}^{n \times n} (A^y = (a_{ij}^y) \in \mathbb{R}^{n \times n})$ is the coupling matrix of sub-network X(Y), and if a connection exists between node $i$ and node $j(i \neq j)$, then $a_{ij}^x(a_{ij}^y) = 1$, otherwise $a_{ij}^x(a_{ij}^y) = 0$; the diagonal elements $a_{ii}^x(a_{ii}^y)$ satisfy dissipative condition $a_{ii}^x = -\sum_{j=1, j \neq i}^{N} a_{ij}^x (a_{ii}^y = -\sum_{j=1, j \neq i}^{N} a_{ij}^y)$; $C = (c_{ij}) \in \mathbb{R}^{n \times n}$ is the intercoupling matrix from sub-network Y to sub-network X, that is, if a interdependency exists from node $i$ in sub-network Y to node $j$ in sub-network X, then $c_{ij} = 1$, otherwise $c_{ij} = 0$.

Remark 1. We consider that the coupling functions are different in different sub-networks and $\alpha, \beta, \gamma$ are not equal. These agree with the fact in the real-world.

Remark 2. In this paper, the intercoupling matrix $C$ is not symmetric for unidirectional interdependency and the construction of $C$ follows the law that the node with a high degree in sub-network Y will preferentially depend on the node with a high degree in sub-network X. It is more universal in actual interdependent networks.

To achieve local synchronization, we add controllers $u_i^x$ and $u_i^y$ into two sub-networks respectively. Then Eqs (1) and (2) can be rewritten as

$$\dot{x}_i = f(x_i) + \alpha \sum_{j=1}^{N} a_{ij}^x H(x_j) + u_i^x \quad i = 1, 2, \ldots, N \tag{3}$$

$$\dot{y}_i = g(y_i) + \beta \sum_{j=1}^{N} a_{ij}^y K(y_j) + \gamma \sum_{j=1}^{N} c_{ij}(H(x_j) - K(y_i)) + u_i^y \quad i = 1, 2, \ldots, N \tag{4}$$

Considering the interdependent networks composed of (3) and (4), there are two isolate nodes and the state vectors of them are $s^x(t) \in \mathbb{R}^n$ and $s^y(t) \in \mathbb{R}^n$. $s^x(t)$ and $s^y(t)$ are utilized as reference trajectories for each sub-network, and satisfy

$$\dot{s}^x(t) = f(s^x(t)) \tag{5}$$

$$\dot{s}^y(t) = g(s^y(t)) \tag{6}$$

Remark 3. Asymptotical local synchronization would be achieved in sub-network X and Y respectively if

$$\lim_{t\to\infty}\|x_i - s^x(t)\| = 0, \quad i = 1, 2, \ldots, N \tag{7}$$

$$\lim_{t\to\infty}\|y_i - s^y(t)\| = 0, \quad i = 1, 2, \ldots, N \tag{8}$$

Remark 4. The states of two isolate nodes are totally different and used as reference trajectories for sub-network X and Y, respectively. So, the asymptotical synchronization in sub-network X is not the same as the one in sub-network Y. We call the asymptotical synchronization in the interdependent networks as asymptotical local synchronization.

In order to design appropriate $u_i^x$ and $u_i^y$, we need the following assumptions and lemma.

Assumption 1. Let $F(t) = Df(s^x(t)) = [f_{ij}(t)]_{n\times n}$ be Jacobian matrix of function $f(s^x(t))$ on $s^x(t)$. $F = (f_{ij})_{n\times n} \in \mathbb{R}^{n\times n}$, and $f_{ij}$ is the maximum value of $f_{ij}(t)$ $(t \in \mathbb{R})$; Let $G(t) = Dg(s^y(t)) = [g_{ij}(t)]_{n\times n}$ be Jacobian matrix of function $g(s^y(t))$ on $s^y(t)$. $G = (g_{ij})_{n\times n} \in \mathbb{R}^{n\times n}$, and $g_{ij}$ is the maximum value of $g_{ij}(t)$ $(t \in \mathbb{R})$.

Assumption 2. Let $B(t) = DH(s^x(t)) = [b_{ij}(t)]_{n\times n}$ be Jacobian matrix of function $H(s^x(t))$ on $s^x(t)$. $B = (b_{ij})_{n\times n} \in \mathbb{R}^{n\times n}$, and $b_{ij}$ is the maximum value of $b_{ij}(t)$ $(t \in \mathbb{R})$; Let $D(t) = DK(s^y(t)) = [k_{ij}(t)]_{n\times n}$ be Jacobian matrix of function $K(s^y(t))$ on $s^y(t)$. $D = (k_{ij})_{n\times n} \in \mathbb{R}^{n\times n}$, and $k_{ij}$ is the maximum value of $k_{ij}(t)$ $(t \in \mathbb{R})$.

Lemma 1. [31] For any matrices $X, Y \in \mathbb{R}^{n\times m}$, if $A^T = A > 0$, $A \in \mathbb{R}^{n\times n}$, then $X^TY + Y^TX \leq X^TAX + Y^TA^{-1}Y$.

## Main results

According to (7) and (8), local synchronization error vectors are defined as

$$e_i^x = x_i - s^x(t) \tag{9}$$

$$e_i^y = y_i - s^y(t) \tag{10}$$

According to (3)–(6), local synchronization error systems can be derived

$$\dot{e}_i^x == f(x_i) - f(s^x(t)) + \alpha \sum_{j=1}^{N} a_{ij}^x H(x_j) + u_i^x \tag{11}$$

$$\dot{e}_i^y = g(y_i) - g(s^y(t)) + \beta \sum_{j=1}^{N} a_{ij}^y K(y_j) + \gamma \sum_{j=1}^{N} c_{ij}(H(x_j) - K(y_i)) + u_i^y \tag{12}$$

### Theorem 1

For the interdependent networks composed of sub-network (3) and (4), if assumptions 1–2 hold, then asymptotical local synchronization could be achieved via controllers

$$u_i^x = -\alpha \sum_{j=1}^{N} a_{ij}^x H(s^x(t)) + d_i^x e_i^x \tag{13}$$

$$u_i^y = -\beta \sum_{j=1}^{N} a_{ij}^y K(s^y(t)) - \gamma \sum_{j=1}^{N} c_{ij} H(s^x(t)) + \gamma \sum_{j=1}^{N} c_{ij} K(s^y(t)) + d_i^y e_i^y \tag{14}$$

$$\dot{d}_i^x = -k_i^x e_i^{xT} e_i^x \tag{15}$$

$$\dot{d}_i^y = -k_i^y e_i^{yT} e_i^y \tag{16}$$

where $d_i^x$ and $d_i^y$ are adaptive laws, $k_i^x$ and $k_i^y$ are feedback gains, $k_i^x > 0$, $k_i^y > 0$, $i = 1,2,\ldots,N$.

Proof. Introduce (13), (14) into error system (11) and (12) respectively and use linearization method.

$$\dot{e}_i^x == F(t)e_i^x + \alpha \sum_{j=1}^{N} a_{ij}^x B(t)e_j^x + d_i^x e_i^x \tag{17}$$

$$\dot{e}_i^y = G(t)e_i^y + \beta \sum_{j=1}^{N} a_{ij}^y D(t)e_j^y + \gamma \sum_{j=1}^{N} c_{ij} B(t)e_j^x - \gamma \sum_{j=1}^{N} c_{ij} D(t)e_i^y + d_i^y e_i^y \tag{18}$$

Let $\lambda_{max}^F$ be the maximum eigenvalue of matrix $(F^T + F)$, $\lambda_{max}^G$ be the maximum eigenvalue of matrix $(G^T + G)$, $\lambda_{max}^B$ be the maximum eigenvalue of matrix $BB^T$, $\lambda_{max}^D$ be the maximum eigenvalue of matrix $DD^T$. Let $\tilde{a}_1^x = \max_{1 \le i \le N}|a_{ij}^x|$, $\tilde{a}_2^x = \max_{1 \le i \le N}|a_{ji}^x|$, $\tilde{a}_1^y = \max_{1 \le i \le N}|a_{ij}^y|$, $\tilde{a}_2^y = \max_{1 \le i \le N}|a_{ji}^y|$, $\tilde{c}_1 = \max_{1 \le i \le N}|c_{ij}|$, $\tilde{c}_2 = \max_{1 \le i \le N}|c_{ji}|$.

Choose the candidate Lyapunov function

$$V(t) = \sum_{i=1}^{N} e_i^{xT} e_i^x + \sum_{i=1}^{N} e_i^{yT} e_i^y + \sum_{i=1}^{N} \frac{(d_i^x + d_*^x)^2}{k_i^x} + \sum_{i=1}^{N} \frac{(d_i^y + d_*^y)^2}{k_i^y} \tag{19}$$

where $d_*^x, d_*^y$ are constant, and satisfy

$d_*^x > (\lambda_{max}^F + \alpha N \tilde{a}_1^x \lambda_{max}^B + \alpha N \tilde{a}_2^x + \gamma N \tilde{c}_2)/2$, $\ d_*^y > (\lambda_{max}^G + \beta N \tilde{a}_1^y \lambda_{max}^D + \gamma N \tilde{c} \lambda_{max}^B + \beta N \tilde{a}_2^y)/2$.

Then, with Eqs (15) and (16), the derivative of $V(t)$ along error system (17) and (18) are obtained as

$$\dot{V}(t) = \sum_{i=1}^{N}(\dot{e}_i^{xT}e_i^x + e_i^{xT}\dot{e}_i^x) + \sum_{i=1}^{N}(\dot{e}_i^{yT}e_i^y + e_i^{yT}\dot{e}_i^y) + 2\sum_{i=1}^{N}\frac{d_i^x + d_*^x}{k_i^x}\dot{d}_i^x + 2\sum_{i=1}^{N}\frac{d_i^y + d_*^y}{k_i^y}\dot{d}_i^y$$

$$
\begin{aligned}
= \ &\sum_{i=1}^{N}\Bigg( e_i^{xT}F^T(t)e_i^x + \alpha\sum_{j=1}^{N}a_{ij}^x e_j^{xT}B^T(t)e_i^x + e_i^{xT}F(t)e_i^x + \alpha\sum_{j=1}^{N}a_{ij}^x e_i^{xT}B(t)e_j^x \\
&+ e_i^{yT}G^T(t)e_i^y + \beta\sum_{j=1}^{N}a_{ij}^y e_j^{yT}D^T(t)e_i^y + \gamma\sum_{j=1}^{N}c_{ij}e_j^{xT}B^T(t)e_i^y \\
&- \gamma\sum_{j=1}^{N}c_{ij}e_i^{yT}D^T(t)e_i^y + e_i^{yT}G(t)e_i^y + \beta\sum_{j=1}^{N}a_{ij}^y e_i^{yT}D(t)e_j^y \\
&+ \gamma\sum_{j=1}^{N}c_{ij}e_i^{yT}B(t)e_j^x - \gamma\sum_{j=1}^{N}c_{ij}e_i^{yT}D(t)e_i^y - 2d_*^x e_i^{xT}e_i^x - 2d_*^y e_i^{yT}e_i^y \Bigg)
\end{aligned}
\tag{20}
$$

According to lemma 1, we can get the results as follows:

$$\alpha\sum_{j=1}^{N}a_{ij}^x e_j^{xT}B^T(t)e_i^x + \alpha\sum_{j=1}^{N}a_{ij}^x e_i^{xT}B(t)e_j^x \leq \alpha\sum_{j=1}^{N}|a_{ji}^x|e_i^{xT}e_i^x + \alpha\sum_{j=1}^{N}|a_{ij}^x|e_i^{xT}B(t)B^T(t)e_i^x \tag{21}$$

$$\beta\sum_{j=1}^{N}a_{ij}^y e_j^{yT}D^T(t)e_i^y + \beta\sum_{j=1}^{N}a_{ij}^y e_i^{yT}D(t)e_j^y \leq \beta\sum_{j=1}^{N}|a_{ji}^y|e_i^{yT}e_i^y + \beta\sum_{j=1}^{N}|a_{ij}^y|e_i^{yT}D(t)D^T(t)e_i^y \tag{22}$$

$$\gamma\sum_{j=1}^{N}c_{ij}e_j^{xT}B^T(t)e_i^y + \gamma\sum_{j=1}^{N}c_{ij}e_i^{yT}B(t)e_j^x \leq \gamma\sum_{j=1}^{N}|c_{ji}|e_i^{xT}e_i^x + \gamma\sum_{j=1}^{N}|c_{ij}|e_i^{yT}B(t)B^T(t)e_i^y \tag{23}$$

Introduce (21)–(23) into (20), and with assumptions 1–2 $\dot{V}(t)$ can be rewritten as

$$
\begin{aligned}
\dot{V}(t) \leq & \sum_{i=1}^{N}\left( e_i^{xT}F^T(t)e_i^x + \alpha\sum_{j=1}^{N}|a_{ji}^x|e_i^{xT}e_i^x + e_i^{xT}F(t)e_i^x \right. \\
& + \alpha\sum_{j=1}^{N}|a_{ij}^x|e_i^{xT}B(t)B^T(t)e_i^x + e_i^{yT}G^T(t)e_i^y + \beta\sum_{j=1}^{N}|a_{ji}^y|e_i^{yT}e_i^y \\
& + \gamma\sum_{j=1}^{N}|c_{ji}|e_i^{xT}e_i^x - \gamma\sum_{j=1}^{N}c_{ij}e_i^{yT}D^T(t)e_i^y + e_i^{yT}G(t)e_i^y \\
& + \beta\sum_{j=1}^{N}|a_{ij}^y|e_i^{yT}D(t)D^T(t)e_i^y + \gamma\sum_{j=1}^{N}|c_{ij}|e_i^{yT}B(t)B^T(t)e_i^y \\
& \left. - \gamma\sum_{j=1}^{N}c_{ij}e_i^{yT}D(t)e_i^y - 2d_*^x e_i^{xT}e_i^x - 2d_*^y e_i^{yT}e_i^y \right) \\
\leq & \sum_{i=1}^{N} e_i^{xT}\left( F^T + F + \alpha\sum_{j=1}^{N}|a_{ij}^x|BB^T + \alpha\sum_{j=1}^{N}|a_{ji}^x| + \gamma\sum_{j=1}^{N}|c_{ji}| - 2d_*^x \right)e_i^x \\
& + \sum_{i=1}^{N} e_i^{yT}\left( G^T + G + \beta\sum_{j=1}^{N}|a_{ij}^y|DD^T + \gamma\sum_{j=1}^{N}|c_{ij}|BB^T + \beta\sum_{j=1}^{N}|a_{ji}^y| \right. \\
& \left. - \gamma\sum_{j=1}^{N}c_{ij}(D^T + D) - 2d_*^y \right)e_i^y \\
< & \left( \lambda_{max}^F + \alpha N\tilde{a}_1^x\lambda_{max}^B + \alpha N\tilde{a}_2^x + \gamma N\tilde{c}_2 - 2d_*^x \right)\sum_{i=1}^{N} e_i^{xT}e_i^x \\
& + \left( \lambda_{max}^G + \beta N\tilde{a}_1^y\lambda_{max}^D + \gamma N\tilde{c}\lambda_{max}^B + \beta N\tilde{a}_2^y - 2d_*^y \right)\sum_{i=1}^{N} e_i^{yT}e_i^y
\end{aligned}
\tag{24}
$$

Note that $d_*^x > (\lambda_{max}^F + \alpha N\tilde{a}_1^x\lambda_{max}^B + \alpha N\tilde{a}_2^x + \gamma N\tilde{c}_2)/2$ and $d_*^y > (\lambda_{max}^G + \beta N\tilde{a}_1^y\lambda_{max}^D + \gamma N\tilde{c}\lambda_{max}^B + \beta N\tilde{a}_2^y)/2$, so $\dot{V}(t) < 0$.
The proof is completed.

## Simulation examples

To verify the theoretical analysis, we construct the interdependent networks composed of two undirected sub-networks. Sub-network X is constructed as WS small world network ($N = 10$, $K = 2$, $P = 0.5$) and sub-network Y is constructed as BA scale free network ($N = 10$, $m_0 = 3$, $m = 2$). The characteristics of sub-network X(Y) are as follows: the average path length is 1.6 (1.62), the clustering coefficient is 0.39(0.76) and the average degree is 4(3.4). The nodes in sub-network Y unidirectionally depend on the nodes in sub-network X with one-to-one mode. The construction of interlinks follows the law that the node with a high degree in sub-network Y will preferentially depend on the node with a high degree in sub-network X. The

coupling matrices $A^x$, $A^y$ and the intercoupling matrix $C$ are as below:

$$A^x = \begin{bmatrix} -4 & 1 & 1 & 0 & 0 & 0 & 0 & 0 & 1 & 1 \\ 1 & -4 & 0 & 1 & 0 & 0 & 0 & 0 & 1 & 1 \\ 1 & 0 & -4 & 1 & 1 & 1 & 0 & 0 & 0 & 0 \\ 0 & 1 & 1 & -4 & 1 & 1 & 0 & 0 & 0 & 0 \\ 0 & 0 & 1 & 1 & -4 & 0 & 1 & 0 & 0 & 1 \\ 0 & 0 & 1 & 1 & 0 & -3 & 0 & 0 & 1 & 0 \\ 0 & 0 & 0 & 0 & 1 & 0 & -3 & 1 & 1 & 0 \\ 0 & 0 & 0 & 0 & 0 & 0 & 1 & -3 & 1 & 1 \\ 1 & 1 & 0 & 0 & 0 & 1 & 1 & 1 & -6 & 1 \\ 1 & 1 & 0 & 0 & 1 & 0 & 0 & 1 & 1 & -5 \end{bmatrix} \qquad (25)$$

$$A^y = \begin{bmatrix} -9 & 1 & 1 & 1 & 1 & 1 & 1 & 1 & 1 & 1 \\ 1 & -6 & 1 & 1 & 0 & 1 & 0 & 1 & 0 & 1 \\ 1 & 1 & -3 & 0 & 1 & 0 & 0 & 0 & 0 & 0 \\ 1 & 1 & 0 & -2 & 0 & 0 & 0 & 0 & 0 & 0 \\ 1 & 0 & 1 & 0 & -3 & 0 & 1 & 0 & 0 & 0 \\ 1 & 1 & 0 & 0 & 0 & -3 & 0 & 0 & 1 & 0 \\ 1 & 0 & 0 & 0 & 1 & 0 & -2 & 0 & 0 & 0 \\ 1 & 1 & 0 & 0 & 0 & 0 & 0 & -2 & 0 & 0 \\ 1 & 0 & 0 & 0 & 0 & 1 & 0 & 0 & -2 & 0 \\ 1 & 1 & 0 & 0 & 0 & 0 & 0 & 0 & 0 & -2 \end{bmatrix} \qquad (26)$$

$$C = \begin{bmatrix} 0 & 0 & 0 & 0 & 0 & 0 & 0 & 0 & 1 & 0 \\ 0 & 0 & 0 & 0 & 0 & 0 & 0 & 0 & 0 & 1 \\ 1 & 0 & 0 & 0 & 0 & 0 & 0 & 0 & 0 & 0 \\ 0 & 0 & 0 & 1 & 0 & 0 & 0 & 0 & 0 & 0 \\ 0 & 1 & 0 & 0 & 0 & 0 & 0 & 0 & 0 & 0 \\ 0 & 0 & 1 & 0 & 0 & 0 & 0 & 0 & 0 & 0 \\ 0 & 0 & 0 & 0 & 1 & 0 & 0 & 0 & 0 & 0 \\ 0 & 0 & 0 & 0 & 0 & 1 & 0 & 0 & 0 & 0 \\ 0 & 0 & 0 & 0 & 0 & 0 & 1 & 0 & 0 & 0 \\ 0 & 0 & 0 & 0 & 0 & 0 & 0 & 1 & 0 & 0 \end{bmatrix} \qquad (27)$$

Each node in sub-network X is a Lorenz system

$$\begin{cases} \dot{x}_{i1} = -a_1 x_{i1} + a_1 x_{i2} \\ \dot{x}_{i2} = b_1 x_{i1} - x_{i2} - x_{i1} x_{i3} \qquad i = 1, 2, \cdots, 10 \\ \dot{x}_{i3} = x_{i1} x_{i2} - c_1 x_{i3} \end{cases} \qquad (28)$$

where $a_1 = 10$, $b_1 = 28$, $c_1 = 8/3$, and Lorenz system is in chaos.

Each node in sub-network Y is a Rössler system

$$\begin{cases} \dot{y}_{i1} = -y_{i2} - y_{i3} \\ \dot{y}_{i2} = y_{i1} + a_2 y_{i2} \qquad\qquad i = 1, 2, \cdots, 10 \\ \dot{y}_{i3} = b_2 + y_{i3}(y_{i1} - c_2) \end{cases} \qquad (29)$$

where $a_2 = 0.1$, $b_2 = 0.1$, $c_2 = 14$, and Rössler system is in chaos.

The coupling strength $\alpha = \beta = 0.01$, and the intercoupling strength $\gamma = 0.03$. the coupling functions $H(x_i) = [sinx_{i1}, cosx_{i2}, x_{i3}]^T$, $K(y_i) = [tany_{i1}, y_{i2}, y_{i3}]^T$, $i = 1, 2, \cdots, 10$. The other initial values are as below: $s^x(0) = [0, -20, -5]^T$, $s^y(0) = [10, 0, 0]^T$, $x_i(0) = [i, -i, i]^T$, $y_i(0) = [0.3i, 0.3i, -i]^T$, $d_i^x(0) = 0.1i$, $d_i^y(0) = 0.2i$, $k_i^x = 0.1i$, $k_i^y = i$, $i = 1, 2, \cdots, 10$.

All of our simulations are run in MATLAB R2016a and the time step is 0.01.

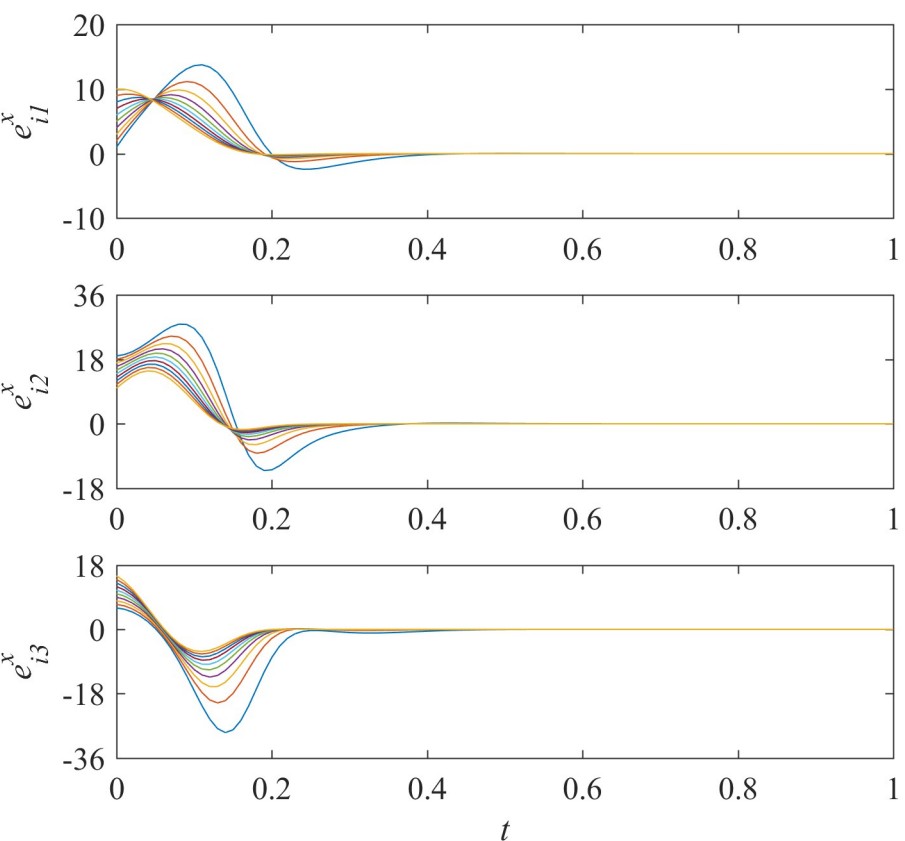

**Fig 1. States of error system between the nodes and the isolate node in sub-network X (Eq (3)).**

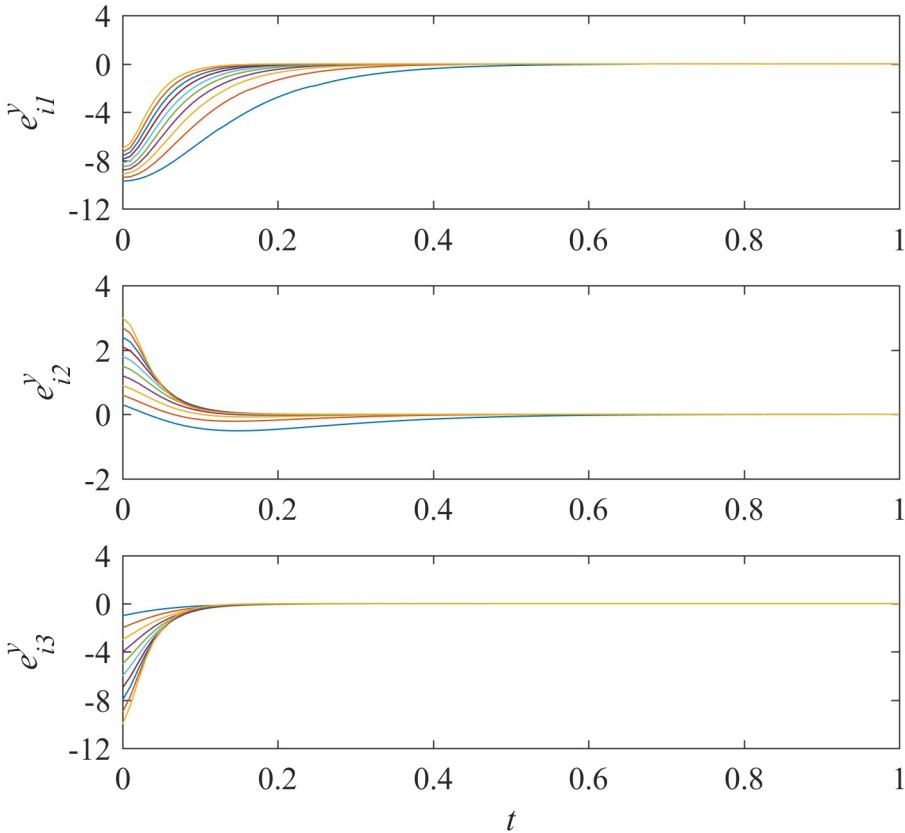

**Fig 2. States of error system between the nodes and the isolate node in sub-network Y (Eq (4)).**

## Example 1

Add controllers (13)–(16) into interdependent networks (3) and (4) according to Theorem 1. The states of error systems are shown in Figs 1 and 2. We can find that the states of error systems tend to zero quickly. That is to say, the proposed control scheme works, and asymptotically local synchronization in each sub-network is achieved respectively.

When asymptotically local synchronization was achieved in each sub-network, the values of adaptive laws tend to be stable. The trajectories of adaptive laws are shown in Figs 3 and 4.

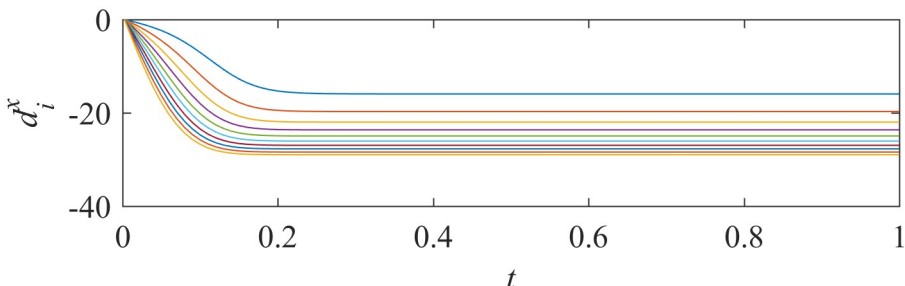

**Fig 3. Trajectory of adaptive laws of sub-network X (Eq (3)).**

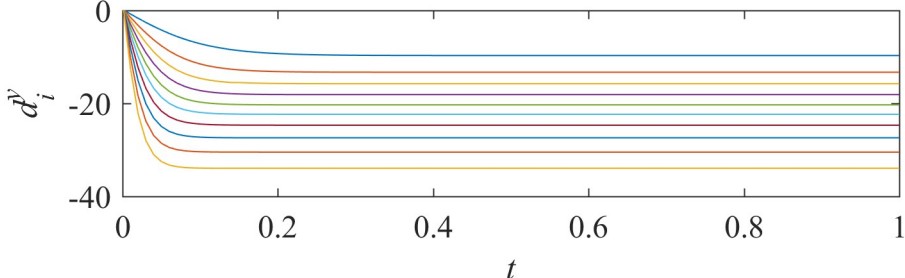

**Fig 4. Trajectory of adaptive laws of sub-network Y (Eq (4)).**

### Example 2

Sub-network Y unidirectionally depends on sub-network X. If all controllers in sub-network Y were out of order, would asymptotically local synchronization in sub-network X still be achieved? The nodes in sub-network Y have no influence on the nodes in sub-network X as the interdependencies are unidirectional. So, the synchronization in sub-network X should be maintained intuitively. We run the simulations and the results are shown in Figs 5 and 6.

The results shown in Fig 5 represents that the synchronization in sub-network X is still achieved while the controllers in sub-network Y do not work. Meanwhile, Fig 6 shows that the states of error system in sub-network Y are oscillatory as expected.

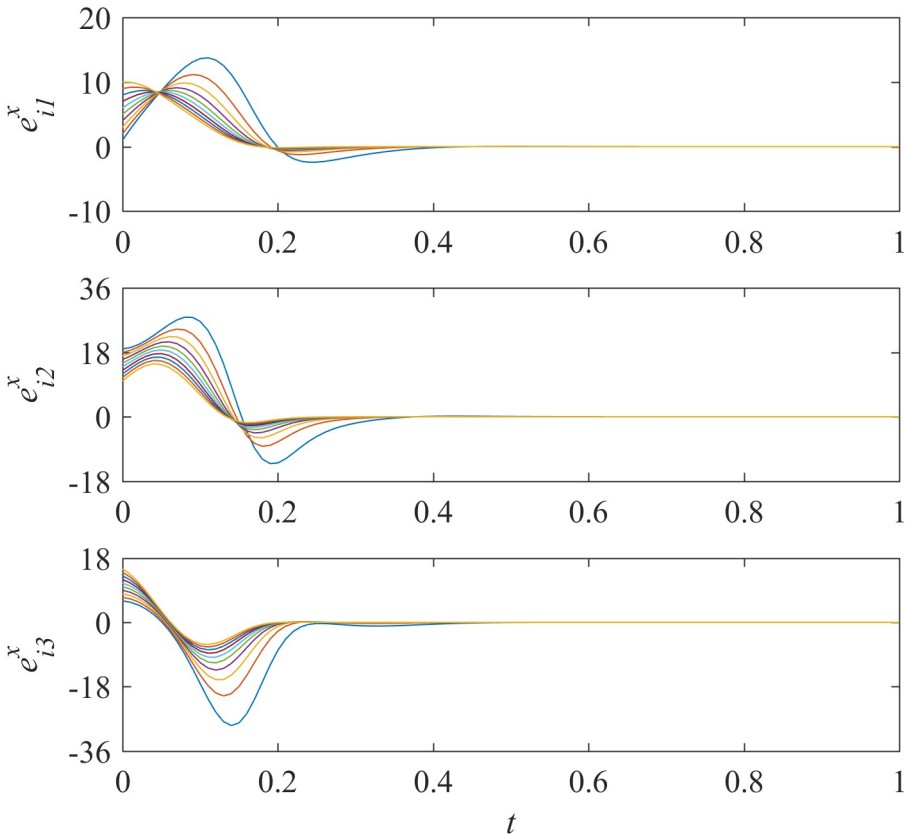

**Fig 5. States of error system between the nodes and the isolate node in sub-network X (Eq (3)).**

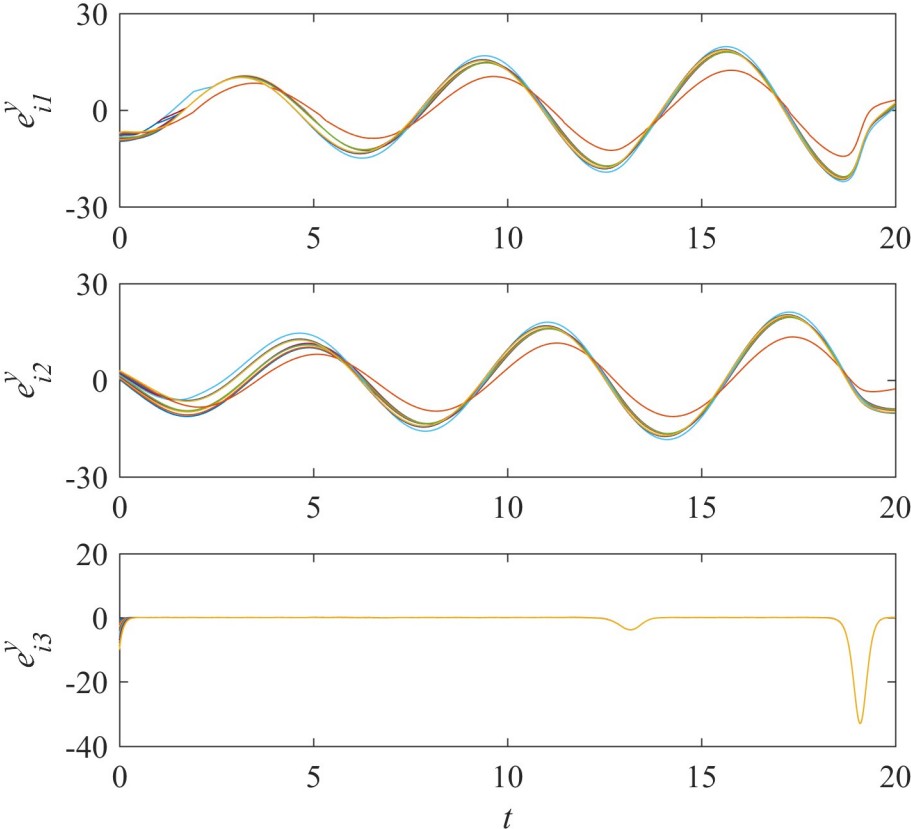

**Fig 6. States of error system between the nodes and the isolate node in sub-network Y (Eq (2)).**

## Example 3

Here we suppose all controllers in sub-network X do not work. This is just opposite to example 2. Simulations are run and the results are given in Figs 7 and 8.

From Fig 7, we can see that the states of error system in sub-network X fall into oscillation. That is to say, the states of the nodes in sub-network X are in chaos. Because of unidirectional interdependency, this would have a great influence on the nodes in sub-network Y and should result in loss of the synchronization in sub-network Y. But on the contrary, the results in Fig 8 indicate that the synchronization in sub-network Y is still achieved. This means that the influence is decreased by using the proposed control scheme. The synchronization can be guaranteed in sub-network Y, even though the nodes in sub-network X are in chaos.

## Example 4

In the interdependent networks, the failure of one or more nodes in one sub-network will result in the cascading failure of the corresponding nodes in another sub-network due to the existence of the interdependency. So it is worth observing whether the synchronization implemented by our controllers is robust against the disturbances.

In our work, the interdependency is unidirectional and one-to-one mode. For simplicity, the connectivity of each sub-network is not consideration and two cases, the nodes in sub-network X or Y are out of work early, are studied in the simulations. Firstly, some nodes in sub-network X or Y are chosen randomly to be out of work. Then the connections between the

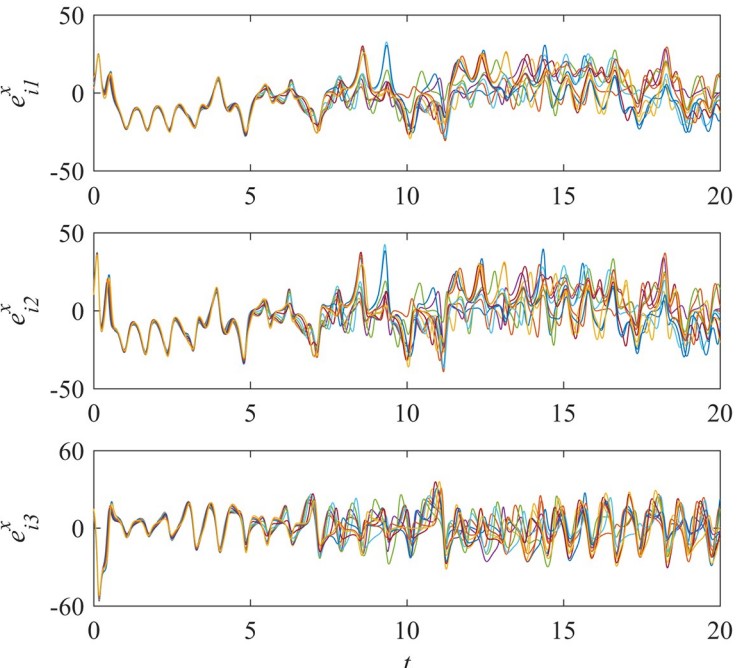

**Fig 7. States of error system between the nodes and the isolate node in sub-network X (Eq (1)).**

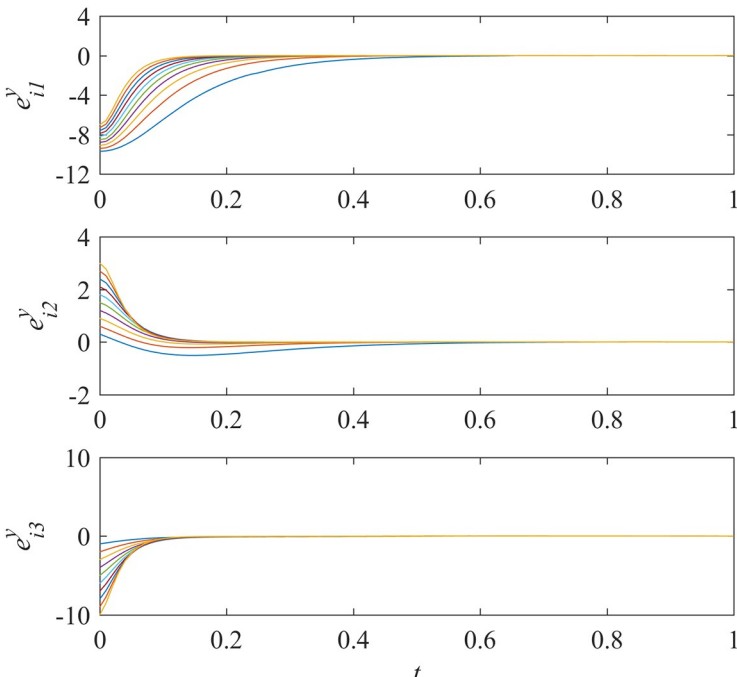

**Fig 8. States of error system between the nodes and the isolate node in sub-network Y (Eq (4)).**

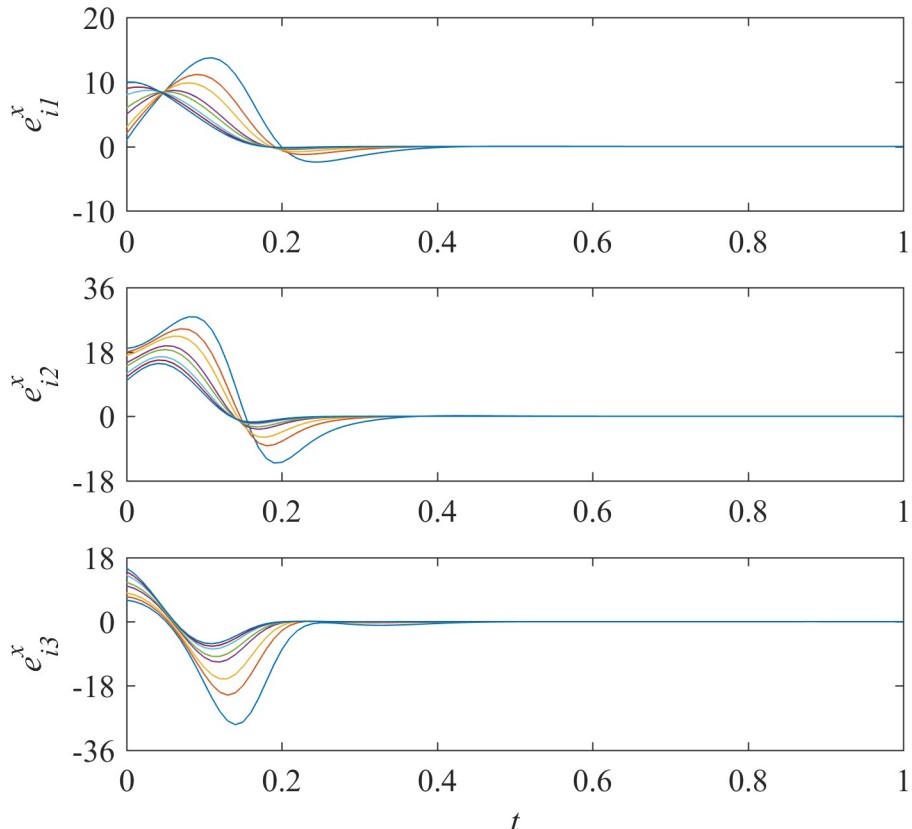

**Fig 9. States of error system between the remaining nodes and the isolate node in sub-network X while node 4 and 7 are chosen to be out of work.**

invalid nodes and their neighbor are deleted. Secondly, if the failed nodes belong to sub-network X, the cascading failure starts, i.e., the corresponding nodes in sub-network Y will also be invalid due to the interdependency. But if the failed nodes belong to sub-network Y, the cascading failure will not start in sub-network X for the interdependency is unidirectional. Finally, the cascading failure results in the deletion of the connections between the failed nodes and their neighbor in sub-network Y.

It is found in our simulations that whether different nodes or different numbers of nodes are chosen early in sub-network X or Y, the results are similar. So only two simulation results are given in Figs 9 and 10 by choosing node 4 and 7 in sub-network X to be out of work early, and in Figs 11 and 12 by choosing node 1 and 6 in sub-network Y early.

From Figs 9–12, it is shown that the synchronization by using our controllers could be retained when some nodes in sub-network X or Y are out of work. Whether the cascading failure happens or not, our control scheme is still effective, which shows the robustness of the proposed method.

## Conclusion

In this article, we design adaptive controllers to achieve asymptotically local synchronization in unidirectional interdependent networks. In the proposed model, different coupling strengths, different coupling functions, intercoupling strength, and intercoupling matrix are considered to agree with the fact in real-world. The feasibility of the control scheme is proved

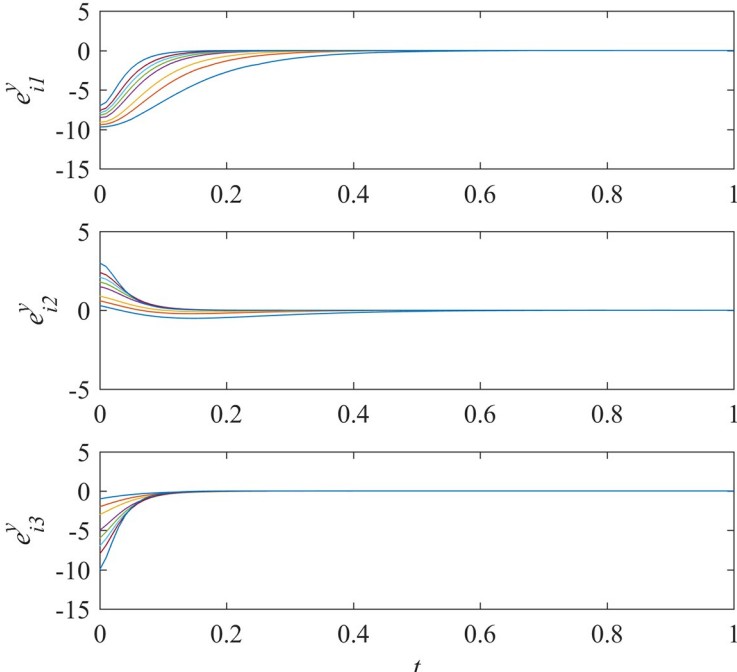

**Fig 10. States of error system between the remaining nodes and the isolate node in sub-network Y with the failed node 4 and 9 for the cascading failure.**

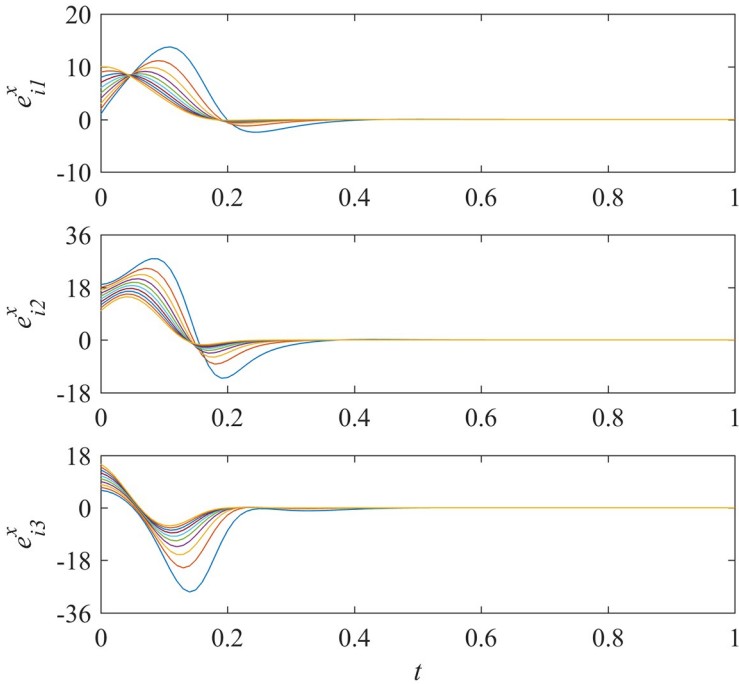

**Fig 11. States of error system between the nodes and the isolate node in sub-network X which is immune to the cascading failure.**

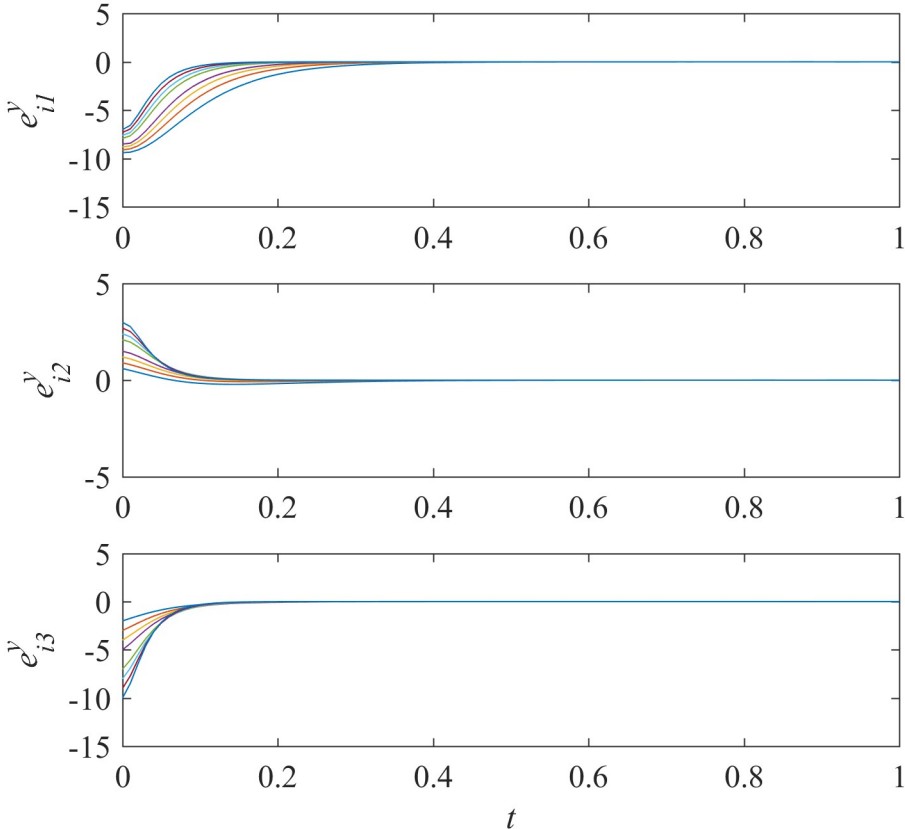

**Fig 12. States of error system between the remaining nodes and the isolate node in sub-network Y while node 1 and 6 are chosen to be out of work.**

theoretically by using Lyapunov stability theory and verified by simulations in MATLAB. The numerical results show that asymptotically local synchronization in unidirectional interdependent networks can be achieved quickly via the adaptive controllers. Furthermore, we find that the synchronization in one sub-network can be achieved by using our control scheme, even if the failure of the controllers exists in the other sub-network. This indicates that the influence can be decreased to a certain extent. Also, the effectiveness of our control scheme is verified while the cascading failure occurs, i.e., the synchronization of the remaining nodes in each sub-network can be retained.

Our work enriches the research contents of the synchronization in the interdependent networks. Further investigations on the synchronization in interdependent networks are needed to promote the deeper research of complex networks.

## Supporting information

**S1 Appendix. A Hurwitz matrix *A* is given which satisfies the condition.**
(DOCX)

## Acknowledgments

The authors are grateful to the anonymous referees for their helpful comments that improved this paper.

## Author Contributions

**Conceptualization:** Zilin Gao, Weimin Luo, Aizhong Shen.

**Data curation:** Zilin Gao, Weimin Luo, Aizhong Shen.

**Formal analysis:** Zilin Gao, Weimin Luo, Aizhong Shen.

**Funding acquisition:** Zilin Gao.

**Investigation:** Zilin Gao, Weimin Luo, Aizhong Shen.

**Methodology:** Zilin Gao, Weimin Luo, Aizhong Shen.

**Resources:** Zilin Gao, Weimin Luo, Aizhong Shen.

**Software:** Zilin Gao, Weimin Luo, Aizhong Shen.

**Validation:** Zilin Gao, Weimin Luo, Aizhong Shen.

**Visualization:** Zilin Gao, Weimin Luo, Aizhong Shen.

**Writing – original draft:** Zilin Gao, Weimin Luo, Aizhong Shen.

**Writing – review & editing:** Zilin Gao, Weimin Luo, Aizhong Shen.

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
