## [Decision Letter · Decision Letter 0]

8 Nov 2021

PONE-D-21-30585Asymptotically local synchronization in interdependent networks with unidirectional interlinksPLOS ONE

Dear Dr. shen,

Thank you for submitting your manuscript to PLOS ONE. After careful consideration, we feel that it has merit but does not fully meet PLOS ONE’s publication criteria as it currently stands. Therefore, we invite you to submit a revised version of the manuscript that addresses the points raised during the review process.

ACADEMIC EDITOR: For stability analysis, Lyapunov function is presented in Eq.(19), it seems that no gains are considered in the error variables e(x,i), it is important to clarify how to select k_i.

We look forward to receiving your revised manuscript.

Kind regards,

Jun Ma, Dr.

Academic Editor

PLOS ONE

Whilst you may use any professional scientific editing service of your choice, PLOS has partnered with both American Journal Experts (AJE) and Editage to provide discounted services to PLOS authors. Both organizations have experience helping authors meet PLOS guidelines and can provide language editing, translation, manuscript formatting, and figure formatting to ensure your manuscript meets our submission guidelines. To take advantage of our partnership with AJE, visit the AJE website (http://aje.com/go/plos) for a 15% discount off AJE services. To take advantage of our partnership with Editage, visit the Editage website (www.editage.com) and enter referral code PLOSEDIT for a 15% discount off Editage services.  If the PLOS editorial team finds any language issues in text that either AJE or Editage has edited, the service provider will re-edit the text for free.

A clean copy of the edited manuscript (uploaded as the new *manuscript* file)”"

3. PLOS requires an ORCID iD for the corresponding author in Editorial Manager on papers submitted after December 6th, 2016. Please ensure that you have an ORCID iD and that it is validated in Editorial Manager. To do this, go to ‘Update my Information’ (in the upper left-hand corner of the main menu), and click on the Fetch/Validate link next to the ORCID field. This will take you to the ORCID site and allow you to create a new iD or authenticate a pre-existing iD in Editorial Manager. Please see the following video for instructions on linking an ORCID iD to your Editorial Manager account: https://www.youtube.com/watch?v=_xcclfuvtxQ. 

“This research has been partly supported by the Scientific Research Project of Chongqing Three Gorges University (19QN07).”

We note that you have provided additional information within the Acknowledgements Section. Please note that funding information should not appear in the Acknowledgments section or other areas of your manuscript. We will only publish funding information present in the Funding Statement section of the online submission form.

“ZG

19QN07

Scientific Research Project of Chongqing Three Gorges University

www.sanxiau.edu.cn

Reviewers comments:

 Reviewer #1: Luo et al. studied the asymptotically local synchronization in interdependent networks with unidirectional interlinks. The topic of the manuscript is of interest. Some interesting results can be obtained. For instance, using the control scheme proposed, synchronization can be realized at least in a sub-network. But, some required revisions should be appropriately addressed as follow:

1. The authors must clarify what is asymptotically local synchronization.

2. The authors should clarify the time step in numerical simulations. Whether using the different time step and initial values cause the obvious differences in the main results.

3. Sub-network Y unidirectionally depends on sub-network X. Whether he global synchronization in the whole network (including X and Y) can be implemented under the condition that controllers are only implanted sub-network X. I think, this is more interesting for unidirectional interdependent networks.

4. Whether the results obtained are robust against the disturbances which exists in the inter-coupling matrix, such as deleting an interlink.

Reviewer #2: This paper studies synchronization in interdependent networks. Since interlinks are not always symmetric in interdependent networks, the president work focuses on the control scheme for synchronization in unidirectional interdependent networks. The feasibility of the proposed control scheme is proved theoretically by the authors using Lyapunov stability theory and verified by simulations. The authors also find that synchronization can be maintained in one sub-network by utilizing their control scheme while the nodes in the other sub-network are in chaos. These results appear to be mathematically well-founded and may be of potential interest in understanding synchronization in interdependent networks. The paper could be improved in several ways:

1. The presentation of the paper might be improved if the rather long proof of Theorem 1 was presented in an appendix, with the just basic idea of the proof being explained in the body of the paper.

2. Although the paper is quite understandable, it would be helpful if the authors could try to improve the English in the paper.

3. Have the authors considered applying their methods to study synchronization with adaptive couplings? For example, it would be interesting if the authors could comment on possible applications of their approach to the adaptive synchronization studied in Chakravartula et al. (2017) Emergence of local synchronization in neuronal networks with adaptive couplings. PLoS ONE 12(6): e0178975.

---

## [Author Response · Author response to Decision Letter 0]

16 Feb 2022

Question from ACADEMIC EDITOR Jun Ma

For stability analysis, Lyapunov function is presented in Eq.(19), it seems that no gains are considered in the error variables e(x,i), it is important to clarify how to select k_i.

Answer:

Dear Dr. Ma, k_i^x and k_i^y are feedback gains, k_i^x>0, k_i^y>0, i=1,2,…,N. They are undetermined and adjustable, and appear at first time in Eq. (15) and (16). In some cases,k_i must be large enough to accelerate the process of synchronization. But in our simulations, we find that both k_i^x and k_i^ywith the values of appropriate size can satisfy the requirements of synchronization. As a result, we let k_i^x=0.1i, k_i^y=i, i=1,2,⋯,10in our simulations.

We appreciate your advice very much.

In addition, we change the order of the authors in our manuscript. All of us admit that the contribution of Dr. Gao is the most important although our contributions are equal. Please accept this modification.

Journal requirements

Answer:

We have ensured that our manuscript meets PLOS ONE's style requirements.

Whilst you may use any professional scientific editing service of your choice, PLOS has partnered with both American Journal Experts (AJE) and Editage to provide discounted services to PLOS authors. Both organizations have experience helping authors meet PLOS guidelines and can provide language editing, translation, manuscript formatting, and figure formatting to ensure your manuscript meets our submission guidelines. To take advantage of our partnership with AJE, visit the AJE website (http://aje.com/go/plos) for a 15% discount off AJE services. To take advantage of our partnership with Editage, visit the Editage website (www.editage.com) and enter referral code PLOSEDIT for a 15% discount off Editage services. If the PLOS editorial team finds any language issues in text that either AJE or Editage has edited, the service provider will re-edit the text for free.

A clean copy of the edited manuscript (uploaded as the new *manuscript* file)”"

Answer:

We have thoroughly copyedited our manuscript for language usage, spelling, and grammar. We do this by ourselves as carefully as possible. If you are still dissatisfied with our manuscript, please inform us.

3. PLOS requires an ORCID iD for the corresponding author in Editorial Manager on papers submitted after December 6th, 2016. Please ensure that you have an ORCID iD and that it is validated in Editorial Manager. To do this, go to ‘Update my Information’ (in the upper left-hand corner of the main menu), and click on the Fetch/Validate link next to the ORCID field. This will take you to the ORCID site and allow you to create a new iD or authenticate a pre-existing iD in Editorial Manager. Please see the following video for instructions on linking an ORCID iD to your Editorial Manager account: https://www.youtube.com/watch?v=_xcclfuvtxQ. 

Answer:

The first author (Zilin Gao) has visited the ORCID site and gotten an ORCID iD: 0000-0001-6214-3312.

The corresponding author (Aizhong Shen) has visited the ORCID site and gotten an ORCID iD: 0000-0001-5158-8615.

“This research has been partly supported by the Scientific Research Project of Chongqing Three Gorges University (19QN07).”

We note that you have provided additional information within the Acknowledgements Section. Please note that funding information should not appear in the Acknowledgments section or other areas of your manuscript. We will only publish funding information present in the Funding Statement section of the online submission form.

Please remove any funding-related text from the manuscript and let us know how you would like to update your Funding Statement. Currently, your Funding Statement reads as follows:“ZG19QN07 Scientific Research Project of Chongqing Three Gorges Universitywww.sanxiau.edu.cn

Answer:

We have removed the funding-related text from the manuscript and updated our Funding Statement in cover letter.

Answer:

 All parameters in our simulations have been given in the manuscript. If anyone wants to rerun the simulations to verify our results, just use these parameters in MATLAB. The version of MATLAB is R2016a and the time step is 0.01, which is also stated in the manuscript.

Comments from Reviewer #1

Luo et al. studied the asymptotically local synchronization in interdependent networks with unidirectional interlinks. The topic of the manuscript is of interest. Some interesting results can be obtained. For instance, using the control scheme proposed, synchronization can be realized at least in a sub-network. But, some required revisions should be appropriately addressed as follow:

1. The authors must clarify what is asymptotically local synchronization.

Answer:

The interdependent networks are composed of two sub-networks at least. As so far, the global synchronization in the interdependent networks is too difficult to be achieved. So, we focus on the local synchronization, that is, the states of the nodes in the same sub-network can be synchronized. In our work, we utilize two isolate nodes as reference trajectories for the nodes in each sub-network. If the difference between the isolate node and the nodes in a sub-network could tend to be zero as time goes by, then we consider that the asymptotically synchronization in the sub-network would be achieved. From an overall perspective, the asymptotically local synchronization in the interdependent networks is achieved when the asymptotically synchronization in each sub-network is achieved.

 We have revised the manuscript on page 7 and added Remark 4 to explain what is asymptotically local synchronization. The concept of asymptotically local synchronization is given more clearly than before.

2. The authors should clarify the time step in numerical simulations. Whether using the different time step and initial values cause the obvious differences in the main results.

Answer:

The time step in our simulations is 0.01. This has been added into the manuscript on page 14.

3. Sub-network Y unidirectionally depends on sub-network X. Whether the global synchronization in the whole network (including X and Y) can be implemented under the condition that controllers are only implanted sub-network X. I think, this is more interesting for unidirectional interdependent networks.

Answer:

This advice is so precious that we have discussed for several times. We also agree with that the advice is more interesting and is similar to our previous ideas (our current study). However, this is different from the control scheme proposed in our manuscript. We achieve asymptotically local synchronization in interdependent networks by adding two different types of the controllers in two sub-networks, respectively. And we find the synchronization could be maintained in one sub-network while the controllers in the other sub-network do not work. To realize the advice, the controllers must be redesigned to satisfy the requirements.

So, we decide to take the advice as another research issue. It would be an interesting topic for future research.

4. Whether the results obtained are robust against the disturbances which exists in the inter-coupling matrix, such as deleting an interlink.

Answer:

Thank you for your advice. In the interdependent networks, the inter-coupling matrix represents the interdependent relations between the nodes which belong to different sub-networks. The disturbances should occur when the node does not work and as a result, the coupling matrices and the inter-coupling will change. So, simply deleting the interlink is inappropriate.

In our work, the interdependency is unidirectional and one-to-one mode. Here we still assume sub-network Y depends on sub-network X, i.e., the states of the nodes in the sub-network X have influences on the states of the nodes in the sub-network Y. For simplicity, only two cases are considered:

1) One or more nodes in the sub-network X are out of work. This will result in the failure of the corresponding nodes in the sub-network Y. The connectivity of each sub-network is affected. Can these remaining nodes in each sub-network retain synchronization?

2) One or more nodes in the sub-network Y are out of work. This has no impact on the nodes in the sub-network X for the interdependency is unidirectional. The synchronization of the nodes in the sub-network X will not vary with no doubt. But the connectivity of the sub-network Y perhaps is destroyed. Can these remaining nodes in the sub-network Y retain synchronization?

We add example 4 in the manuscript and two cases are discussed in detail. Thank you again for your valuable comments.

Comments from Reviewer #2

This paper studies synchronization in interdependent networks. Since interlinks are not always symmetric in interdependent networks, the president work focuses on the control scheme for synchronization in unidirectional interdependent networks. The feasibility of the proposed control scheme is proved theoretically by the authors using Lyapunov stability theory and verified by simulations. The authors also find that synchronization can be maintained in one sub-network by utilizing their control scheme while the nodes in the other sub-network are in chaos. These results appear to be mathematically well-founded and may be of potential interest in understanding synchronization in interdependent networks. The paper could be improved in several ways:

1. The presentation of the paper might be improved if the rather long proof of Theorem 1 was presented in an appendix, with the just basic idea of the proof being explained in the body of the paper.

Answer:

 We have created a new word document named as Appendix 1. In the document, all equations in the manuscript are shown and the proof of Theorem 1 is presented in details. We simplify the proof in the manuscript to make it more explained.

2. Although the paper is quite understandable, it would be helpful if the authors could try to improve the English in the paper.

Answer:

 We apologize for our poor English writing ability. We have tried our best to copyedit the manuscript and wish to get your approval.

3. Have the authors considered applying their methods to study synchronization with adaptive couplings? For example, it would be interesting if the authors could comment on possible applications of their approach to the adaptive synchronization studied in Chakravartula et al. (2017) Emergence of local synchronization in neuronal networks with adaptive couplings. PLoS ONE 12(6): e0178975.

Answer:

The interdependent networks in the manuscript are indeed a nonlinear complex system with time-invariant couplings. The coupling in one sub-network or the intercoupling between sub-networks is constant. And the strength of the coupling or the intercoupling do not change.

According to your recommendation, we have downloaded the paper and read it carefully. Chakravartula et al. study adaptively coupled neuronal networks composed of Hindmarsh-Rose neurons. The coupling between two neurons is determined dynamically by the states of the neurons. In the paper, Eq. (4) and (7) are given as below:

█(x ˙_i=y_i-x_i^3+bx_i^2-z_i+I+∑_(j=1)^N▒〖A_ij k_ij (x_j-x_i ) 〗#(4) )

█(k ˙_ij=k_ij [αe^(-β(x_i-x_j )^2 )-γ(k_ij+1) ]#(7) )

Chakravartula et al. put forth an idea that the coupling strengthsk_ij is not fix but adaptive, and assume k_ij vary according to Eq (7). After analyzing some features of the dynamics of Eq (7), they run simulations. They find the networks naturally produce both permanent and transient synchronization of local clusters of neurons, and the emergence of a power-law spectrum in the deterministic system appears to be a consequence of a novel form of self-organized criticality.

However, the paper is different from our work. Our point is to add controllers in the interdependent networks with the unidirectional interlink to achieve asymptotically local synchronization. When our goal is achieved, we find the synchronization could be retained in one sub-network while the controllers are out of work in the other sub-network, especially in the sub-network which depends on the other sub-network. But the point of Chakravartula et al. is to observe the dynamics of the local synchronization inoneadaptively coupled neuronal network, in which the coupling between two neurons is determined dynamically by the states of the neurons.

Although the difference exists, we appreciate your advice. It provides some novel ideas for our further study. The content of the manuscript is supplemented on page 4 and the paper you recommended is used as one reference article in the manuscript.

---

## [Decision Letter · Decision Letter 1]

19 Apr 2022

Asymptotically local synchronization in interdependent networks with unidirectional interlinks

PONE-D-21-30585R1

Dear Dr. shen,

We’re pleased to inform you that your manuscript has been judged scientifically suitable for publication and will be formally accepted for publication once it meets all outstanding technical requirements.

Kind regards,

Jun Ma, Dr.

Academic Editor

PLOS ONE

Reviewer #1: The presentation of this version has been greatly improved. The problems I care have been addressed.

Reviewer #2: The authors have carefully revised their paper taking into account the reviewers’ comments and it is now suitable for publication.

---

## [Editor Report · Acceptance letter]

25 Apr 2022

PONE-D-21-30585R1 

Asymptotically local synchronization in interdependent networks with unidirectional interlinks 

Dear Dr. shen:

I'm pleased to inform you that your manuscript has been deemed suitable for publication in PLOS ONE. Congratulations! Your manuscript is now with our production department. 

Kind regards, 

on behalf of

Dr. and Pro. Jun Ma 

Academic Editor

PLOS ONE